# 3-Hydroxy-3-Methylglutaric Acid Disrupts Brain Bioenergetics, Redox Homeostasis, and Mitochondrial Dynamics and Affects Neurodevelopment in Neonatal Wistar Rats

**DOI:** 10.3390/biomedicines12071563

**Published:** 2024-07-15

**Authors:** Josyane de Andrade Silveira, Manuela Bianchin Marcuzzo, Jaqueline Santana da Rosa, Nathalia Simon Kist, Chrístofer Ian Hernandez Hoffmann, Andrey Soares Carvalho, Rafael Teixeira Ribeiro, André Quincozes-Santos, Carlos Alexandre Netto, Moacir Wajner, Guilhian Leipnitz

**Affiliations:** 1Programa de Pós-Graduação em Ciências Biológicas: Bioquímica, Universidade Federal do Rio Grande do Sul, Rua Ramiro Barcelos, 2600-Anexo, Porto Alegre 90035-003, RS, Brazil; josysilveira94@gmail.com (J.d.A.S.); manuelamarcuzzo9@gmail.com (M.B.M.); jaquelinesantanarosa@gmail.com (J.S.d.R.); nathaliakist@gmail.com (N.S.K.); christoferianhh@gmail.com (C.I.H.H.); andreyscarvalho@outlook.com (A.S.C.); rafaelrated@hotmail.com (R.T.R.); andrequincozes@ufrgs.br (A.Q.-S.); netto@gabinete.ufrgs.br (C.A.N.); mwajner@ufrgs.br (M.W.); 2Departamento de Bioquímica, Instituto de Ciências Básicas da Saúde, Universidade Federal do Rio Grande do Sul, Rua Ramiro Barcelos, 2600-Anexo, Porto Alegre 90035-003, RS, Brazil; 3Programa de Pós-Graduação em Neurociências, Universidade Federal do Rio Grande do Sul, Rua Ramiro Barcelos, 2600-Anexo, Porto Alegre 90035-003, RS, Brazil; 4Programa de Pós-Graduação em Ciências Biológicas: Fisiologia, Universidade Federal do Rio Grande do Sul, Rua Ramiro Barcelos, 2600-Anexo, Porto Alegre 90035-003, RS, Brazil; 5Serviço de Genética Médica, Hospital de Clínicas de Porto Alegre, Rua Ramiro Barcelos, 2350, Porto Alegre 90035-903, RS, Brazil

**Keywords:** 3-hydroxy-3-methylglutaric acidemia, bioenergetics, oxidative stress, neurodevelopment, brain

## Abstract

3-Hydroxy-3-methylglutaric acidemia (HMGA) is a neurometabolic inherited disorder characterized by the predominant accumulation of 3-hydroxy-3-methylglutaric acid (HMG) in the brain and biological fluids of patients. Symptoms often appear in the first year of life and include mainly neurological manifestations. The neuropathophysiology is not fully elucidated, so we investigated the effects of intracerebroventricular administration of HMG on redox and bioenergetic homeostasis in the cerebral cortex and striatum of neonatal rats. Neurodevelopment parameters were also evaluated. HMG decreased the activity of glutathione reductase (GR) and increased catalase (CAT) in the cerebral cortex. In the striatum, HMG reduced the activities of superoxide dismutase, glutathione peroxidase, CAT, GR, glutathione S-transferase, and glucose-6-phosphate dehydrogenase. Regarding bioenergetics, HMG decreased the activities of succinate dehydrogenase and respiratory chain complexes II–III and IV in the cortex. HMG also decreased the activities of citrate synthase and succinate dehydrogenase, as well as complex IV in the striatum. HMG further increased DRP1 levels in the cortex, indicating mitochondrial fission. Finally, we found that the HMG-injected animals showed impaired performance in all sensorimotor tests examined. Our findings provide evidence that HMG causes oxidative stress, bioenergetic dysfunction, and neurodevelopmental changes in neonatal rats, which may explain the neuropathophysiology of HMGA.

## 1. Introduction

3-Hydroxy-3-methylglutaric acidemia (HMGA) is a neurometabolic recessive disorder caused by the deficient activity of 3-hydroxy-3-methylglutaryl-coenzyme A lyase (HL) affecting ketone body synthesis and leucine catabolism. Patients present with a predominant accumulation of 3-hydroxy-3-methylglutaric acid (HMG) in the brain and biological fluids [1]. Other organic acids are also found to be elevated but at lower levels, such as 3-methylglutaric, 3-methylglutaconic, and 3-hydroxyisovaleric acids [2]. The levels of all these acids further increase during episodes of acute metabolic decompensation along with hypoglycemia and metabolic acidosis, which usually occurs in the neonatal period [3,4]. The diagnosis is based on the measurement of the urinary levels of organic acids and the acylcarnitine profile in the blood. The number of individuals with confirmed HMGA is estimated to be approximately 200 worldwide [5].

The symptomatology of HMGA is similar to that of Reye syndrome and includes severe hypo/non-ketotic hypoglycemia, vomiting, metabolic acidosis, developmental delay, impaired vigilance, hypotonia, lethargy, convulsions, and coma. Macrocephalia is also observed [5,6,7]. The clinical signs usually appear or worsen after metabolic decompensation, which is characterized by a dramatic elevation in organic acid levels, as described above [1,2]. MRI findings show a high frequency of cerebral atrophy, and basal ganglia and white matter abnormalities [5].

The pathophysiology of HMGA is not fully established. Nevertheless, in vitro and ex vivo studies from our group have shown that HMG increases reactive oxygen species levels, disturbs the antioxidant system, and induces oxidative damage to different biomolecules in the brain of young rats. Further results revealed that HMG induces oxidative stress and also disrupts energy metabolism in other tissues of rats, such as liver and heart [8,9]. In the brain of 1-day-old rats, HMG further increased mitochondrial fusion and induced neural damage [10].

Given that crises of acute metabolic decompensation in HMGA patients usually occur in the first year of life [5,11], we evaluated the ex vivo effects of a single intracerebroventricular administration of HMG, the major accumulating metabolite in HMGA, on redox homeostasis, bioenergetics, and mitochondrial dynamics in neonatal rat cerebral cortex and striatum, the brain structures that are most injured in HMGA patients [2,5,6]. Furthermore, we investigated the possible in vivo disturbances in neonatal neurodevelopment triggered by HMG through the assessment of sensorimotor reflexes. It is worth noting that 7-day-old Wistar rats show a similar level of brain maturation to preterm human newborns [12]. Thus, the experimental design proposed in this study mimics the clinical outcomes of HMGA in the first months of life.

## 2. Materials and Methods

### 2.1. Animals and Reagents

Newborn Wistar rats (7-day-old) were obtained from the Centro de Reprodução e Experimentação de Animais de Laboratório (CREAL), ICBS, UFRGS, Brazil. The dams and pups were maintained on a 12:12 h light/dark cycle (lights on 7 a.m.–7 p.m.) in a constant-temperature (22 ± 1 °C) colony room, with free access to water and 20% (*w*/*w*) protein commercial chow (SUPRA, São Leopoldo, RS, Brazil). The experiments were performed following the national animal rights legislation (Law 11794/2008) and the experimental procedures reported in the Guide for the Care and Use of Laboratory Animals (National Institutes of Health, publication no. 80-23, revised 2011) and Directive 2010/63/EU. The Committee on the Ethical Use of Animals of UFRGS approved the protocol (project numbers 32807 and 44267). All chemicals were purchased from Sigma-Aldrich Co. (St. Louis, MO, USA).

### 2.2. Ex Vivo Experiments

#### 2.2.1. HMG Administration

A total of forty-four male Wistar rats (7-day-old) and four dams were used in the study. The neonatal rats (11–13 g) were randomly assigned to two groups: control or test. Appendix A shows the distribution of the animals used to determine the parameters.

Neonatal rats were slightly anesthetized with a cotton pad embedded in isoflurane and gently restrained, and they received a single intracerebroventricular (icv) administration (fourth ventricle) of PBS (vehicle; control) or HMG (1 μmol/g; test) into the Magna cisterna fourth ventricle as previously described [13,14]. HMG solution was prepared in PBS immediately before injection, adjusting the pH to 7.4. The rats were kept on a heating pad for 15–20 min after the injection to recover and then returned to the dam in their home cage. The rats were euthanized 6 h after injection, and the cerebral cortex (temporal, parietal, and occipital cortices) and striatum were dissected and prepared for the evaluation of bioenergetics and redox homeostasis parameters. This approach aims to mimic an acute metabolic decompensation crisis that commonly occurs in HMGA. The content of specific proteins was not evaluated by Western blotting in the striatum because this cerebral structure is too small, and the protein concentration of the samples is not adequate for this technique.

Different littermates were used to assess neurodevelopment 48 h after injection (postnatal day 9—PND9). After the assessment, the pups were returned to their dam and euthanized on the same day.

#### 2.2.2. Antioxidant Defenses

The cerebral cortex and the striatum were homogenized (1:10 *w*/*v*) in 20 mM sodium phosphate buffer, pH 7.4, containing 140 mM KCl. After centrifugation at 750× *g* for 10 min (4 °C), the supernatants were separated and used for the investigation of antioxidant defenses. GSH concentrations were determined by fluorimetry based on the reaction of GSH with o-phthaldialdehyde generating a product that was read at 350 (excitation) and 420 (emission) nm [15]. Superoxide dismutase (SOD) activity was measured indirectly through the autooxidation of pyrogallol, which was read at 420 nm [16]. Catalase (CAT) activity was assessed by monitoring the decomposition of hydrogen peroxide at 240 nm [17]. Glutathione peroxidase (GPx) activity was evaluated by monitoring NADPH oxidation at 340 nm [18]. Glutathione S-transferase (GST) activity was determined by accompanying dinitrophenyl-S-glutathione formation from 1-chloro-2,4-dinitrobenzene at 340 nm [19]. Glucose-6-phosphate dehydrogenase (G6PDH) activity was evaluated by following NADPH generation at 340 nm [20]. Glutathione reductase (GR) activity was measured by monitoring NADPH oxidation at 340 nm [21]. All these parameters were measured using a SpectraMax M5 microplate reader (San Jose, CA, USA). These methods are described in detail in Grings et al. (2017) [22].

### 2.3. Bioenergetics

The cerebral cortex and the striatum were homogenized (1:10 *w*/*v*) in SET buffer (250 mM sucrose, 2.0 mM EDTA, and 10 mM Trizma base), pH 7.4. The homogenates were centrifuged at 800× *g* for 10 min at 4 °C, and the supernatants obtained were used for the assessment of respiratory chain complex and citric acid cycle (CAC) enzyme activities. All these parameters were measured using a SpectraMax M5 microplate reader (San Jose, CA, USA).

The supernatants were frozen and thawed three times before the determination of the activities of the respiratory chain complexes. Complex II activity was assessed by following the reduction of DCIP, while the activity of complexes II–III was measured by monitoring cytochrome *c* reduction [23]. Complex IV activity was assessed by following cytochrome *c* oxidation [24].

As for the CAC enzymes, citrate synthase (CS) activity was determined by measuring 5,5-dithio-bis (2-nitrobenzoic acid) (DTNB) reduction at 412 nm (Shepherd and Garland 1969). Succinate dehydrogenase (SDH) activity was determined by measuring DCIP reduction at 600 nm [23]. Malate dehydrogenase (MDH) activity was determined by following the reduction in NADH fluorescence at 366 (excitation) and 450 (emission) nm [25].

### 2.4. Western Blotting

The cerebral cortex was homogenized (1:5 *w*/*v*) in RIPA buffer containing 1 mM sodium orthovanadate and 1% protease inhibitor cocktail. The homogenate was centrifuged at 3631× *g* for 10 min at 4 °C, and the protein concentration was determined by the method of Lowry et al. (1951) [26]. The assay is detailed in da Rosa-Junior et al. (2019) [27]. The following primary antibodies were used: anti-mitofusin 1 (MFN1; 1:750, ab57602, Abcam, Cambridge, UK; RRID: AB_2142624), anti-dynamin-related protein 1 (DRP1; 1:1000, ab56788, Abcam, Cambridge, UK; RRID: AB_941306), optic atrophy 1 (OPA-1, 1:1000, #80471, D6U6N, Cell Signaling Technologies, Danvers, MA, USA; RRID:AB_2734117), and anti-β-actin (1:10,000, #4967, Sigma-Aldrich, MO, USA; RRID: AB_330288). Mouse anti-rabbit IgG-HRP (1:5000; SC-2357, Santa Cruz Biotechnology, TX, USA; RRID: AB_628497) and mouse IgG kappa binding protein (1:5000; SC-516102, Santa Cruz Biotechnology, Dallas, TX, USA; RRID: AB_2687626) were used as secondary antibodies. Membrane images were cropped to highlight the bands of interest. Immunoreactivity was detected by employing a chemiluminescence (ECL) detection substrate (Clarity Western ECL Substrate, Bio-Rad Laboratories Inc., Hercules, CA, USA), and chemiluminescence signals were captured using a charge-coupled device (CCD) camera (ImageQuant™ LAS 4000, GE Health Care, Piscataway, NJ, USA).

### 2.5. Neurodevelopmental Reflex Assessment

Neurodevelopmental reflex was evaluated on PND9 by conducting negative geotaxis, righting reflex, gait, cliff avoidance, hindlimb suspension, and forelimb grasping tests. Each reflex was performed in triplicate between 9 a.m. and 12 p.m. by a blinded researcher. In the negative geotaxis test, the latency (with a maximum time of 30 s) was assessed for the animals to move 180 degrees on a 45-degree inclined plane. To assess the righting reflex, the pups were placed in a supine position and the latency (seconds) of the animals to turn with all four paws on the ground was measured. The locomotion was evaluated using the gait test, in which the animals were placed in the center of a 15 cm circle and the latency to leave the circle was calculated. In the cliff avoidance test, the pups were placed on a ledge 30 cm above the ground, and the protective responses were scored 0 when the pup did not move or fell from the ledge, 1 when there was minimal effort to get off the cliff but the limbs were kept on the cliff, and 2 when there was movement away from the cliff. The hindlimb suspension test was used to assess neuromuscular function. The animals were placed in a 50 mL laboratory glass beaker and the posture of their hind legs was observed. The posture was classified according to the following criteria: A score of 0 indicated that the animals were unable to maintain their hindlimbs on the edge of the container or a permanent lowering of the tail, 1 indicated that the animals exhibited muscle weakness during the test, 2 indicated that the hind limbs were frequently in contact, 3 indicated that the hindlimbs were close but not touching, and 4 represented an adequate distance between the hindlimbs and an extended tail. The forelimb grasping reflex was quantified using the following scoring system: 0 for no grasping, 1 for grasping with just one paw, and 2 for grasping with both paws. Further details regarding the methodology employed in this study can be found in Ribeiro et al. [28].

### 2.6. Protein Determination

Protein content was measured according to Lowry et al. (1951) [26].

### 2.7. Statistical Analysis

Results are presented as mean ± standard deviation (SD). The mean was used for statistical analysis. No blinding procedures were used for the experiments. The sample size (n) was initially estimated with Minitab 16 software for experiments with two groups. We assumed a target power of 0.8, an SD of 10%, and a difference of 35%, as used for previous studies [16,28]. In some experiments, we used a slightly different n because they were not performed on the same day and the littermates provided by the Central Animal House had a variable number of animals [16,28]. In addition, most neurodevelopment parameters had *n* = 6, but negative geotaxis and cliff avoidance tests had *n* = 5 because one HMG-treated rat could not perform these tests.

Data were analyzed by Student’s *t*-test for unpaired samples when F was significant. The normality of data was assessed by the Shapiro–Wilk test. When data were not normal, the Mann–Whitney test was used. No test for outliers was performed. *p* < 0.05 was rated significant. Statistical Package for the Social Sciences (SPSS) software 21 (Armonk, NY, USA) was used.

## 3. Results

### 3.1. HMG Impairs Enzymatic Antioxidant Defenses in the Neonatal Brain

In the first set of experiments, we evaluated the effects of HMG administration on the antioxidant defenses in the cerebral cortex and the striatum of neonatal rats 6 h after the injection. HMG increased the activity of CAT (t_(8)_ = 0.7761; *p* < 0.01) and reduced the activity of GR (t_(8)_ = 2.421; *p* < 0.05) in the cerebral cortex (Figure 1). However, HMG did not change the activities of SOD, GPx, GST, or G6PDH, or the levels of GSH (Figure 1). Additionally, we found that HMG significantly diminished the activities of SOD (t_(8)_ = 3.069; *p* < 0.05), CAT (t_(8)_ = 3.206; *p* < 0.05), GPx (t_(8)_ = 4.189; *p* < 0.01), GST (t_(8)_ = 3.505; *p* < 0.01), G6PDH (t_(8)_ = 7.106; *p* < 0.001), and GR (t_(8)_ = 2.596; *p* < 0.05) in the striatum, albeit GSH concentrations were not changed (Figure 2).

### 3.2. HMG Disrupts CAC and Mitochondrial Respiratory Chain Functioning in the Neonatal Brain

We further assessed the effects of HMG on the activities of CAC enzymes and respiratory chain complexes. Figure 3 displays that the organic acid reduced the activity of SDH (t_(6)_ = 4.899; *p* < 0.01) but did not modify CS or MDH in the cortex. Moreover, in the cortex, HMG reduced the activities of complexes II–III (t_(6)_ = 6.877; *p* < 0.05) and IV (t_(6)_ = 3.329; *p* < 0.05) but did not change complex II (Figure 3). In the striatum, the administration of HMG decreased the activities of CS (t_(6)_ = 6.460; *p* < 0.05) and SDH (t_(6)_ = 2.151; *p* < 0.01) but did not alter MDH (Figure 3). As for the respiratory chain in the striatum, HMG decreased the activity of complex IV (t_(6)_ = 7.935; *p* < 0.01) (Figure 4). Complexes II and II–III were not changed in the striatum (Figure 4). These findings suggest that high levels of HMG impair the functioning of CAC and the electron transfer in the mitochondrial respiratory chain.

### 3.3. HMG Affects Mitochondrial Dynamics in the Neonatal Cerebral Cortex

In the subsequent experiments, we evaluated the influence of HMG treatment on mitochondrial dynamics in the cerebral cortex since bioenergetic and redox status, which was impaired by HMG, is essential for the maintenance of mitochondrial morphology [29]. HMG markedly increased the content of DRP1 (t_(10)_ = 16.88; *p* < 0.001), indicative of mitochondrial fission. In contrast, MFN1 and OPA1 content was not significantly changed (Figure 5).

### 3.4. HMG Impairs the Neurodevelopment of Neonatal Rats

The assessment of neurodevelopmental reflexes is an important tool for identifying early disturbances in brain maturation [30,31]. Given that HMG caused marked changes in the redox status, bioenergetics, and mitochondrial dynamics in the neonatal rat CNS, we assessed whether this organic acid could alter the neurodevelopment of these animals. Figure 6 shows that the icv infusion of HMG into neonatal rats increased latency in the negative geotaxis test (t_(8)_ = 2.273; *p* < 0.01), suggesting a delayed vestibular response. The organic acid also increased the latency on the righting reflex (U = 0, n1 = n2 = 6, *p* < 0.01) and worsened the performance of the rats in the cliff avoidance test (U = 0, n1 = 6, n2 = 5, *p* < 0.01) (Figure 6). In addition, HMG provoked a reduction in mobility in the gait evaluation (U = 1, n1 = n2 = 6, *p* < 0.01), along with a loss of muscle tone in the hindlimbs (hindlimb suspension test) (t_(5)_ = 2.697; *p* < 0.05) (Figure 6). In contrast, the forelimb grasping reflex was not modified (Figure 6).

## 4. Discussion

HMGA is predominantly characterized by neurological dysfunction and brain abnormalities that may lead a considerable number of patients to death [5]. HMG is the predominant metabolite accumulated in the plasma and tissues of patients and has been shown to elicit neurotoxicity in animal models through impairment of energetics, redox status, and glutamatergic neurotransmission [9,10]. Nevertheless, the consequences of HMG have been poorly studied in the neonatal period, which is a crucial stage of neurodevelopment for appropriate brain maturation [32,33]. Therefore, we evaluated whether the intracerebroventricular administration of HMG could affect enzymatic and non-enzymatic antioxidant defenses, CAC and respiratory chain functioning, mitochondrial dynamics, or neurodevelopment of male rats at this age. Noteworthy, there are no studies reporting gender differences in HMGA.

HMG disturbed the enzymatic antioxidant defenses in both the cerebral cortex and the striatum of rats. Previous reports showed that different enzymatic structures are subjected to ROS-mediated regulation or oxidative attack [34,35,36]. We speculate that in our animal model, the levels of ROS are increased by HMG, thereby disturbing the regulatory mechanisms or causing oxidative damage to different enzymes and possibly leading to their inhibition. This may explain the decreased activities of GR in the cerebral cortex and striatum, as well as of SOD, GPx, CAT, GST, and G6PDH in the striatum, by HMG. Surprisingly, we also verified an increase in the activity of CAT in the cortex caused by HMG. We did not evaluate the gene expression of this enzyme, but HMG likely upregulated it through a specific mechanism that did not occur for the other enzymes [37]. Moreover, the increase in CAT activity might be a cellular response to combat high levels of hydrogen peroxide caused by HMG effects. It is difficult to explain why HMG reduced CAT activity in the striatum but increased it in the cortex. However, it is widely known that the concentrations of iron, which catalyze the formation of reactive species [37], are higher in the striatum than in the cortex [38,39]. Thus, we speculate that the generation of ROS by HMG may occur through an iron-dependent mechanism. This hypothesis may also explain why the activity of many antioxidant enzymes, including CAT, was reduced in the striatum but not in the cortex. As for the lack of changes in GSH levels, it is conceivable that HMG may have increased the synthesis of this antioxidant.

Regarding the bioenergetics parameters, HMG reduced the cortical activity of SDH and the striatal activity of CS and SDH. HMG further reduced the activities of complexes II–III and IV in the cortex and complex IV in the striatum. We have not determined the mechanisms by which HMG inhibited these enzymes and complexes, but it was likely mediated by ROS-induced oxidative attack [37], as previously hypothesized for the antioxidant enzymes. Other mechanisms may be involved, such as decreased mitochondrial biogenesis [40,41,42] or impairment in the targeting of these enzymes to the mitochondria [43]. Interestingly, we recently showed that ethylmalonic acid, an organic acid that accumulates in ethylmalonic encephalopathy, binds to and inhibits α-ketoglutarate dehydrogenase [44], which reinforces that organic acids may directly inhibit bioenergetics-related enzymes. In this context, it should be also noted that complex III is considered an important source of ROS in cells under different pathophysiological conditions [45,46], so the inhibition of this complex may be a main contributor to the redox status changes caused by HMG.

Mitochondrial dynamics encompasses the processes of fusion and fission, which are responsible for the control and regulation of the morphology, function, and distribution of this organelle. Under normal circumstances, there is equilibrium between each of these processes; however, different studies have shown that in pathological states with bioenergetic dysfunction and oxidative stress the mitochondrial dynamics are impaired [47,48,49], which is in line with our findings. HMG increased DRP1 levels in the cerebral cortex, the main protein involved in the regulation of fission, suggesting that this process is stimulated by this acid possibly as an attempt to eliminate dysfunctional mitochondria. In contrast, MFN 1 and OPA1 were not significantly changed. It should be noted that the phosphorylation status and specific isoforms of these proteins were not evaluated, so we cannot rule out that mitochondrial fusion is also affected by HMG.

Biochemical disturbances during the early stages of neurodevelopment can lead to cellular, molecular, and behavioral alterations [28,50,51]. To evaluate whether the alterations in redox homeostasis and bioenergetics induced by HMG could disturb rat neurodevelopment, we first assessed righting reflex, cliff avoidance, and negative geotaxis on PND9. These parameters were determined specifically on PND9 due to the following reasons: (1) to rule out a possible interference of the anesthesia, (2) to give an adequate recovery time after the injection, and (3) to detect the possible neurodevelopment changes with more reliability, since after most insults in animal models such alterations cannot be detected after short periods. It should also be noted that some neurodevelopmental reflexes in rodents start before PND7 [52], i.e., before the intracerebral administration, and that is why we evaluated them only on PND9 and not since their onset. We verified that HMG injection significantly worsened the performance in these parameters, implying an impairment in the sensorimotor response. These responses require appropriate maturation of the cortico-spinal/spinal-cortical projections in order to correctly capture sensory stimuli and execute an appropriate muscular response [28,53]. Moreover, HMG caused marked changes in parameters related to locomotion and muscle tone, reflected by increased gait latency and reduced hindlimb suspension, respectively. Therefore, HMG led to neuromotor impairments in neonatal rats similar to those found in patients affected by HMGA, which may indicate that the biochemical alterations may be strictly related to the pathophysiology of this organic acidemia.

The pathophysiological relevance of our findings must be taken with caution. Although we believe that our chemical model resembles, at least partially, an episode of metabolic decompensation that is commonly observed in individuals with HMGA [2,5], we did not measure the exact concentrations that HMG achieves in the brain of animals. Nevertheless, it should be emphasized that HMG accumulation has been shown in the brain of patients through coupled brain and urine spectroscopy [1], even though no studies have reported the exact levels of this metabolite in the CNS. It should be also noted that the dose used here is similar to that utilized in animal models of other hereditary disorders [28,54,55]. Further studies demonstrated that in some organic acidurias, such as HMGA, the local production and entrapment of dicarboxylates and acyl-CoA ester precursors result in the pronounced accumulation of these compounds in the brain, causing toxic effects that lead to progressive neurodegeneration [56,57]. In such diseases, the intracerebral synthesis and low efflux transport from the brain leading to the entrapment of organic acids have been proposed to participate in brain injury [56,57]. Therefore, although we do not know the brain concentrations of HMG, it is conceivable that this organic acid is found at very high levels in the intracellular milieu.

On the other hand, previous data from our group showed that HMG also caused oxidative stress and impaired the functioning of the citric acid cycle in the cerebral cortex of 1-day-old rats [10]. Although these effects in 1-day-old animals were less pronounced [12], this may be explained by the lower dose of HMG injected (0.5 μmol/g) compared to that used in the present study.

In summary, we showed that HMG causes oxidative stress and bioenergetic dysfunction in the cerebral cortex and striatum of 7-day-old animals. Our work reveals important temporal pathomechanisms underlying the neurological dysfunction of HMGA, since almost half of the patients present with symptoms in the first year of life [5], which corresponds to the age of the animals used in our chemical model [12]. Additionally, we found that the toxic effects led to marked changes in the neurodevelopment of animals. This is consistent with the clinical picture of the disease, which is commonly characterized by developmental delay and hypotonia [5]. Finally, our findings suggest that antioxidants, particularly those targeting mitochondria, and energy substrates could be considered potential therapeutic strategies for HMGA.

## Figures and Tables

**Figure 1 biomedicines-12-01563-f001:**
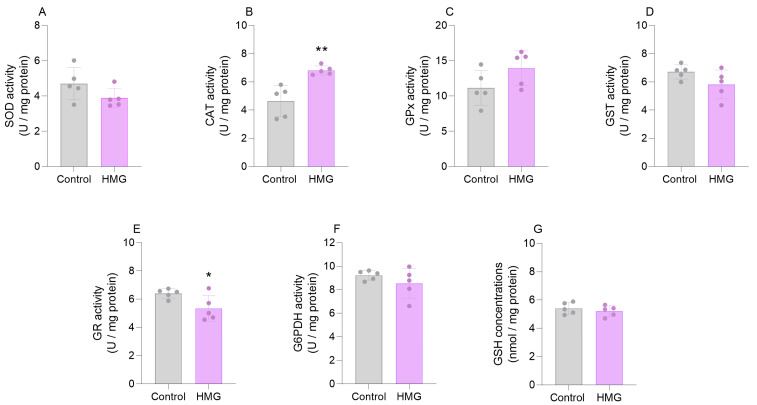
Effects of 3-hydroxy-3-methyglutaric acid (HMG) on antioxidant defenses in the neonatal rat cerebral cortex. Superoxide dismutase (SOD; (**A**)), catalase (CAT; (**B**)), glutathione peroxidase (GPx; (**C**)), glutathione S-transferase (GST; (**D**)), glutathione reductase (GR; (**E**)), and glucose-6-phosphate dehydrogenase (G6PDH; (**F**)) activities and reduced glutathione (GSH) levels (**G**) were measured. Values are means ± SD (*n* = 5). * *p* < 0.05, ** *p* < 0.01, compared to rats receiving PBS (control group) (Student’s *t*-test).

**Figure 2 biomedicines-12-01563-f002:**
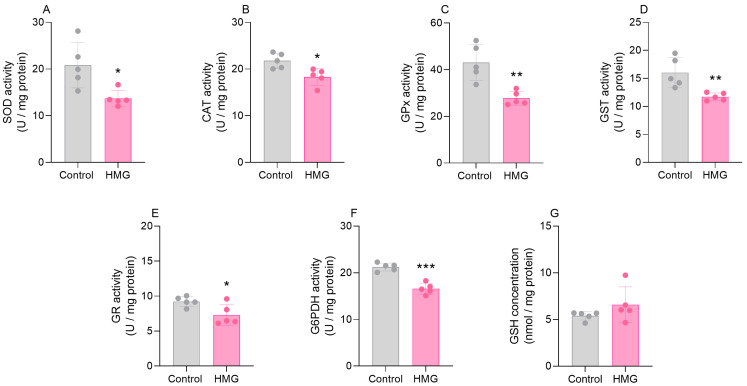
Effects of 3-hydroxy-3-methyglutaric acid (HMG) on antioxidant defenses in the neonatal rat striatum. Superoxide dismutase (SOD; (**A**)), catalase (CAT; (**B**)), glutathione peroxidase (GPx; (**C**)), glutathione S-transferase (GST; (**D**)), glutathione reductase (GR; (**E**)), and glucose-6-phosphate dehydrogenase (G6PDH; (**F**)) activities and reduced glutathione (GSH) levels (**G**) were measured. Values are means ± SD (*n* = 5). * *p* < 0.05, ** *p* < 0.01, *** *p* < 0.001, compared to rats receiving PBS (control group) (Student’s *t*-test).

**Figure 3 biomedicines-12-01563-f003:**
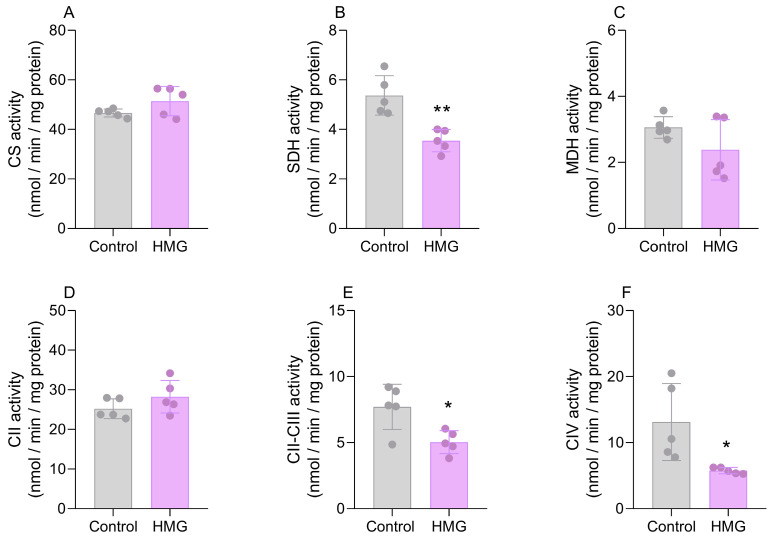
Effects of 3-hydroxy-3-methyglutaric acid (HMG) on the activities of citric acid cycle enzymes and mitochondrial respiratory chain complexes in the neonatal rat cerebral cortex. The activities of the citric acid cycle enzymes citrate synthase (CS; (**A**)), succinate dehydrogenase (SDH; (**B**)), and malate dehydrogenase (MDH; (**C**)), as well as respiratory chain complexes II (CII; (**D**)), II–III (CII-III; (**E**)) and IV (CIV; (**F**)), were measured. Values are means ± SD (*n* = 5). * *p* < 0.05, ** *p* < 0.01, compared to rats receiving PBS (control group) (Student’s *t*-test).

**Figure 4 biomedicines-12-01563-f004:**
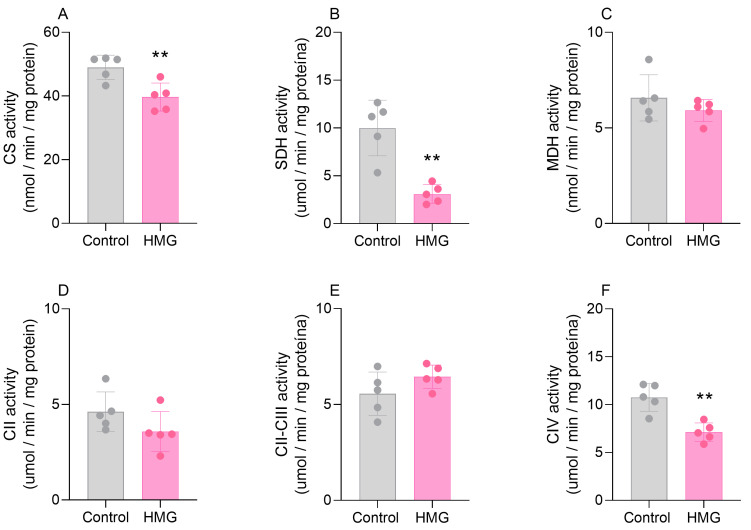
Effects of 3-hydroxy-3-methyglutaric acid (HMG) on the activities of citric acid cycle enzymes and mitochondrial respiratory chain complexes in the neonatal rat striatum. The activities of the citric acid cycle enzymes citrate synthase (CS; (**A**)), succinate dehydrogenase (SDH; (**B**)), and malate dehydrogenase (MDH; (**C**)), as well as the respiratory chain complexes II (CII; (**D**)), II–III (CII-III; (**E**)), and IV (CIV; (**F**)), were measured. Values are means ± SD (*n* = 5). ** *p* < 0.01, compared to rats receiving PBS (control group) (Student’s *t*-test).

**Figure 5 biomedicines-12-01563-f005:**
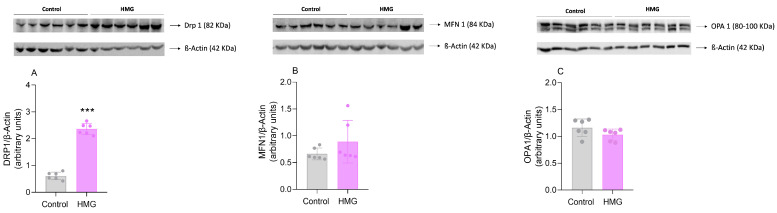
Effects of 3-hydroxy-3-methyglutaric acid (HMG) on the content of DRP1 (**A**), MFN1 (**B**), and OPA1 (**C**) in the neonatal rat cerebral cortex. Representative blots are shown at the top and quantification is at the bottom. Values are means ± SD (*n* = 6). *** *p* < 0.001, compared to rats receiving PBS (control group) (Student’s *t*-test).

**Figure 6 biomedicines-12-01563-f006:**
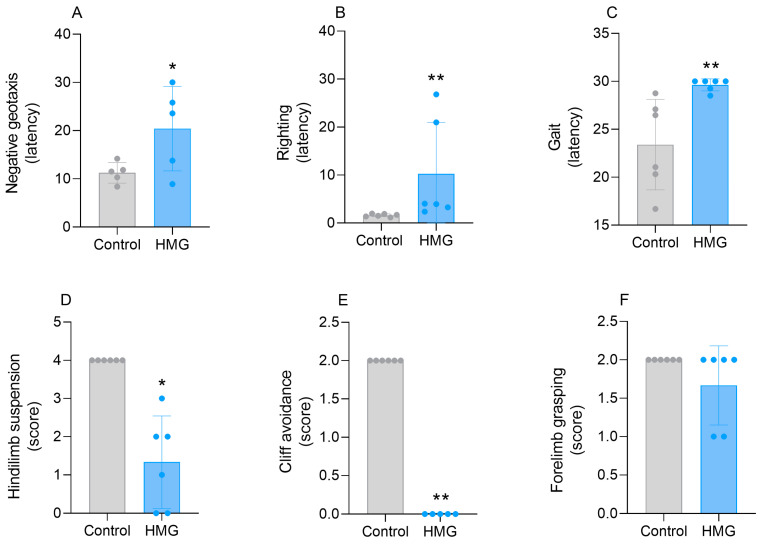
Effects of 3-hydroxy-3-methyglutaric acid (HMG) on neurodevelopmental parameters in neonatal rats. Negative geotaxis (**A**), righting (**B**), gait (**C**), hindlimb suspension (**D**), cliff avoidance (**E**), and forelimb grasping (**F**) were evaluated on postnatal day 9. Values are means ± SD (*n* = 5–6). * *p* < 0.05, ** *p* < 0.01, compared to rats receiving PBS (control group) (Student’s *t*-test or Mann–Whitney test).

## Data Availability

Dataset available on request from the authors due to Privacy.

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
