# Peer review of "3-Hydroxy-3-Methylglutaric Acid Disrupts Brain Bioenergetics, Redox Homeostasis, and Mitochondrial Dynamics and Affects Neurodevelopment in Neonatal Wistar Rats"

_biomedicines, 2024, doi:10.3390/biomedicines12071563_

Round 1

Reviewer 1 Report

Comments and Suggestions for Authors

The authors showed that HMG causes oxidative stress and bioenergetic 348 dysfunction in the cerebral cortex and striatum of 7-day-old animals. Overall, the introduction and methods sections of are detailed, well-structured, and the research hypothesis is clearly outlined. The results are well presented and potential mechanisms are proposed. This article can be published after a few minor corrections.

Only a few minor concerns:

1.    Methods: The animals were injected with HMG at P7 and sacrificed at 6 hours later and behavioral studies conducted at P9. How can this short period of insult simulate the clinical condition of HMGA.

2.    Results: No neuropathology was shown. No data regarding oxidant were shown. Mitochondrial dynamics were limited to 3 proteins. MFN2, FIS1, and MFF were not examined. The phosphorylation status of proteins were not examined.

3.    A few typos or editing errors were seen: line 52, fluid [1] s.; line 89, 22@1; line 177, mean@; line 281, [8,9] [8,9]; line 340, 1 @mol;

4.    Reference style should be consistent. For example, ref 3 and 4 are in different style.

Comments on the Quality of English Language

English is fine.

Author Response

The authors showed that HMG causes oxidative stress and bioenergetic 348 dysfunction in the cerebral cortex and striatum of 7-day-old animals. Overall, the introduction and methods sections of are detailed, well-structured, and the research hypothesis is clearly outlined. The results are well presented and potential mechanisms are proposed. This article can be published after a few minor corrections.

Only a few minor concerns: 

  1. Methods: The animals were injected with HMG at P7 and sacrificed at 6 hours later and behavioral studies conducted at P9. How can this short period of insult simulate the clinical condition of HMGA.

      Response: We cannot determine at present how the time point of 6 hours mimics the clinical condition of HMGA. We selected this time point aiming to evaluate the short-term effects of a single injection of HMG, which resembles an acute episode of metabolic decompensation commonly seen in HMGA. We also followed previous studies showing that other metabolites elicit toxic effects 6 hr after the injection in neonatal rats [Ribeiro et al. Neurotoxicity Research (2023) 41:119–140; Ribeiro et al., Neuroscience (2021) 471:115-132; Zemniaçak et al., Mol Neurobiol (2024) 61(5):2496-2513]. Overall, we believe that the fact that HMG markedly altered redox homeostasis and bioenergetics in such a short period reinforces our hypothesis that HMG accumulation has a key role in the pathophysiology of HMGA.

      We added the following sentence in the Methodology (“HMG administration”):

“This approach aims to mimic an acute metabolic decompensation crisis that commonly occurs in HMGA.”

  1. Results: No neuropathology was shown. No data regarding oxidant were shown. Mitochondrial dynamics were limited to 3 proteins. MFN2, FIS1, and MFF were not examined. The phosphorylation status of proteins were not examined.

      Response: We thank the Reviewer for suggesting that neuropathology should be performed. This would certainly contribute to understanding HMGA pathophysiology and will be performed in a future study. Regarding specifically the determination of other proteins regulating mitochondrial dynamics as well as their phosphorylation status, this is indeed a limitation of our study. However, several reports in the literature have determined only the content of these proteins to investigate mitochondrial dynamics [Lou et al., Arch Toxicol (2013) 87:449–457; Marques-Aleixo et al., Neuroscience 301 (2015) 480–495; Xu et al., Theranostics (2021) 11(2): 522-539], as we have done in our study, suggesting that this is a reasonable approach to analyze mitochondrial dynamics alterations. We inserted the following sentence in the manuscript to raise this issue.

      “In contrast, MFN 1 and OPA1 were not significantly changed. It should be noted that the phosphorylation status and specific isoforms of these proteins were not evaluated so we cannot rule out that mitochondrial fusion is also affected by HMG.”

  1. A few typos or editing errors were seen: line 52, fluid [1] s.; line 89, 22@1; line 177, mean@; line 281, [8,9] [8,9]; line 340, 1 @mol; 

Response: We corrected the typos and editing errors.

  1. Reference style should be consistent. For example, ref 3 and 4 are in different style.

Response: We corrected the reference style.

Reviewer 2 Report

Comments and Suggestions for Authors

I have reviewed the manuscript entitled " 3-Hydroxy-3-methylglutaric acid disrupts brain bioenergetics, redox homeostasis, and mitochondrial dynamics and affects neurodevelopment in neonatal Wistar rats by Josyane de Andrade Silveira and coworkers.

 After reading the manuscript several questions have risen therefore, I suggest the authors to clarify them and to make several corrections to this version before it can be considered to be published.

1)Methodology section

The authors begin stating that they have used 7-day old littermates divided in two groups (control and experimental). However, they do not state the number of dams used to acquire this number of pups. They do not state the sex (female or male) of the pups used within the litter. How were they distributed? Did the litters weight the same at day 7?

How was the experimental procedure done? How did they restrain the pups? Were the 7-day old pups anesthetized? How did the authors inject into the 4th ventricle the HMG?

Another question is: from the 35 animals used in the study, how were they distributed? Basically, two types of experimental procedures were performed: biochemical and behavioral experiments. How were the animals distributed in these experiments? Again, was the sex of the animals considered?

Is the prevalence of this disease equal between males and females?

Why was the 6 h post-injection chosen? Why were the biochemical test performed 6h post injection and the 48 h for behavioral testing? Why weren’t the same times chosen?

As for Neurodevelopmental Reflex Assessment more information should be provided.

In the result section the authors show the “latency” for the negative geotaxis test, the righting and the gating (figure 6). However, the unit to measure of this “latency” is never stated. The same applies to the “score” for hindlimb suspension, cliff avoidance and forelimb grasping.

A very important point when measuring neurodevelopmental reflexes is that the experimenter measures the acquisition of each reflex through the first 21 days of life. A reflex is considered acquired when it is performed for two consecutive days. See the work by: Nguyen AT, Armstrong EA, Yager JY. Neurodevelopmental Reflex Testing in Neonatal Rat Pups. J Vis Exp. 2017 Apr 24;(122):55261. doi: 10.3791/55261. PMID: 28518104; PMCID: PMC5565095.

Please provide more information on how were these milestones measured for validation of the results.

Information on when and how pups were sacrificed after neurodevelopmental reflex assessment is missing.

 Another point that should be addressed is why did the authors choose the cortex and the striatum to measure redox/ antioxidant activity/ mitochondrial activity? Why these structures? Is there a specific reason? Please state.

Lines 248-249 read that MFN1 is not statistically significantly changed but figure 5 shows that it is (*).

In the discussion section the increase found in CAT in the cortex and the decrease found in the striatum needs more detailed explanation.

A reference for lines 332-335 should be provided.

In lines 342-343: “In addition, we showed that HMG also caused oxidative stress and impaired the functioning of citric acid cycle in the cerebral cortex of 1-day-old rats albeit the effects were less pronounced [10]”.

Please rewrite this sentence. It is not clear. It appears that the authors want to discuss their results but they are referring to a work done by other authors.

Lines 346-347: If the authors state that no studies have measured the levels of HMG in the brain of patients at least they should provide information on how this disease is detected in these individuals.

I also suggest that the inclusion of a simple diagram or scheme of the biochemical impaired pathway that leads to HMG accumulation would be useful.

Minor points:

Correct typo mistake line 52

Line 83: I suggest changing the word “moments” for a precise period as the “first year of life” or the perinatal period.

Line 183: please finish the sentence

Line 246: replace the doi for reference number.

Line 281: repetition of references.

Line 340: correct symbol in units.

Best regards!

Comments on the Quality of English Language

Some corrections should by made.

Author Response

I have reviewed the manuscript entitled " 3-Hydroxy-3-methylglutaric acid disrupts brain bioenergetics, redox homeostasis, and mitochondrial dynamics and affects neurodevelopment in neonatal Wistar rats by Josyane de Andrade Silveira and coworkers.

 After reading the manuscript several questions have risen therefore, I suggest the authors to clarify them and to make several corrections to this version before it can be considered to be published.

1)Methodology section

The authors begin stating that they have used 7-day old littermates divided in two groups (control and experimental). However, they do not state the number of dams used to acquire this number of pups. They do not state the sex (female or male) of the pups used within the litter. How were they distributed? Did the litters weight the same at day 7?

Response: We corrected the Methodology section by detailing the information requested, as follows:

“A total of forty-four male Wistar rats (7-day-old) and four dams were used in the study. The neonatal rats (11-13 g) were randomly assigned to two groups: control and test groups. Table S1 shows the distribution of the animals used to determine the parameters.”

How was the experimental procedure done? How did they restrain the pups? Were the 7-day old pups anesthetized? How did the authors inject into the 4th ventricle the HMG?

Response: We altered and added sentences in the text to answer these questions, as follows.

“Neonatal rats were slightly anesthetized with a cotton pad embedded in isoflurane, gently restrained, and received a single intracerebroventricular (icv) administration (fourth ventricle) of PBS (vehicle; control) or HMG (1 μmol/g; test) into the Magna cisterna fourth ventricle as previously described [11,12]. HMG solution was prepared in PBS immediately before injection, adjusting the pH to 7.4. The rats were kept on a heating pad for 15-20 min after the injection to recover and then returned to the dam in their home cage. The rats were euthanized 6 hr after injection, and the cerebral cortex (temporal, parietal, and occipital cortices) and striatum were dissected and prepared for the evaluation of bioenergetics and redox homeostasis parameters.”

Another question is: from the 35 animals used in the study, how were they distributed? Basically, two types of experimental procedures were performed: biochemical and behavioral experiments. How were the animals distributed in these experiments? Again, was the sex of the animals considered?

Response: We actually used a total of 44 animals, which were distributed in the experiments according to the table below. We apologize for the mistake.

Supplementary Table 1. Number of animals used in each experimental group for evaluating the biochemical parameters and neurodevelopment

Parameters

Number of rats in Control group

Number of rats in test group

Antioxidant enzyme activities and reduced glutathione levels

5

5

Citric acid cycle enzyme and respiratory chain complex activities

5

5

Western blotting

6

6

Neurodevelopment markers

6

6

 As mentioned in the previous answer, we used male rats.

Is the prevalence of this disease equal between males and females?

Response: Unfortunately, no study has reported whether there is a difference in the prevalence among the sexes.

Why was the 6 h post-injection chosen? Why were the biochemical test performed 6h post injection and the 48 h for behavioral testing? Why weren’t the same times chosen?

Response: The time point of 6 hr was chosen because we aimed to investigate the short-term effects of HMG administration, which resembles an episode of metabolic decompensation that is commonly seen in HMGA patients. In addition, previous studies from our group showed that other metabolites accumulating in inborn errors of metabolism elicit toxic effects at this or similar periods [Ribeiro et al. Neurotoxicity Research (2023) 41:119–140; Ribeiro et al., Neuroscience (2021) 471:115-132; Zemniaçak et al., Mol Neurobiol (2024) 61(5):2496-2513].

As for neurodevelopment, the following reasons led us to evaluate it 48 hr after the injection: 1) we wanted to rule out a possible interference of the anesthesia on the markers; 2) to give a recovery time after HMG administration; 3) after most insults in animal models, changes in neurodevelopment cannot be detected after short periods.

The following was inserted in the Discussion.

“These parameters were determined specifically on PND9 due to the following reasons: 1) to rule out a possible interference of the anesthesia; 2) to give an adequate recovery time after the injection; 3) to detect the possible neurodevelopment changes with more reliability since after most insults in animal models, such alterations cannot be detected after short periods. It should be also noted that some neurodevelopmental reflexes in rodents start before PND7 [45], i.e. before the intracerebral administration, and that is why we evaluated them only on PND9 and not since their onset.

As for Neurodevelopmental Reflex Assessment more information should be provided.

In the result section the authors show the “latency” for the negative geotaxis test, the righting and the gating (figure 6). However, the unit to measure of this “latency” is never stated. The same applies to the “score” for hindlimb suspension, cliff avoidance and forelimb grasping.

Response: We did not provide a complete description of each test due to possible plagiarism since, as described in the text, all details can be found in Ribeiro et al [23]. However, we added the sentences below to guide the reader.

“The time (latency measured in s) to perform negative geotaxis test, righting and gating was recorded and the maximum latency was assigned when the task was not completed. In turn, scores were given by an experienced researcher for hindlimb suspension, cliff avoidance, and forelimb grasping.”

A very important point when measuring neurodevelopmental reflexes is that the experimenter measures the acquisition of each reflex through the first 21 days of life. A reflex is considered acquired when it is performed for two consecutive days. See the work by: Nguyen AT, Armstrong EA, Yager JY. Neurodevelopmental Reflex Testing in Neonatal Rat Pups. J Vis Exp. 2017 Apr 24;(122):55261. doi: 10.3791/55261. PMID: 28518104; PMCID: PMC5565095.

Please provide more information on how were these milestones measured for validation of the results.

Response: We thank the Reviewer for this important question. There is a number of reasons behind our decision to assess the impact exclusively on DPN9. Firstly, we should mention that our experimental approach consisted of the icv injection of HMG on PND7 because this is an age at which the rats have a brain development similar to a preterm human newborn [Sizonenko et al., Pediatr Res (2003) 54(2):263-9). This was already stated in the last paragraph of the article’s Introduction. Secondly, although it is known that the daily assessment protocol to determine the exact day of acquisition of neurodevelopmental reflexes is a widely used method for early neurobehavioral assessment, the onset of several of the neurodevelopmental reflexes performed in our study occurs before PND7. Therefore, our approach did not allow for verifying the day of reflex acquisition in our experimental design. In this sense, Nguyen et al. (2017) (the manuscript mentioned by the Reviewer) showed that the forelimb grasp and righting reflexes appeared on PND3, cliff avoidance on PND4, and gait on PND6. In addition, Feather-Schussler et al. [J Vis Exp (2016) 117:53569] postulated that the average age for the appearance of the negative geotaxis reflex in rodents is PND7. Conversely, there are scientific publications that, rather than determining the day on which the reflex appears, assessed possible sensorimotor impairments on a specific day [Tassinari et al., Neuroscience (2020) 10:448:191-205; Fabres et al., Metab Brain Dis (2022) 37(7):2315-2329; Machado et al., Neurotox Res (2023) 41(6):526-545]. Therefore, we believe that the reasons mentioned, and the references cited validate our present results.

As described in the previous answer, the following part was added to the Discussion.

“These parameters were determined specifically on PND9 due to the following reasons: 1) to rule out a possible interference of the anesthesia; 2) to give an adequate recovery time after the injection; 3) to detect the possible neurodevelopment changes with more reliability since after most insults in animal models, such alterations cannot be detected after short periods. It should be also noted that some neurodevelopmental reflexes in rodents start before PND7 [45], i.e. before the intracerebral administration, and that is why we evaluated them only on PND9 and not since their onset.”

Information on when and how pups were sacrificed after neurodevelopmental reflex assessment is missing. 

Response: To answer this, we added the following sentences in the manuscript.

“Different littermates were used for assessing neurodevelopment 48 hr after injection (postnatal day 9-PND9). After the assessment, pups were returned to their dam and euthanized on the same day.”

 Another point that should be addressed is why did the authors choose the cortex and the striatum to measure redox/ antioxidant activity/ mitochondrial activity? Why these structures? Is there a specific reason? Please state.

Response: We evaluated the effects of HMG on cerebral cortex and striatum because they are often affected in patients with HMGA. So, we altered the last paragraph of the Introduction to state that, as follows.

Given that episodes of acute metabolic decompensation in HMGA patients usually occur in the first year of life [5], we evaluated the ex vivo effects of a single intracerebroventricular administration of HMG, the major accumulating metabolite in HMGA, on redox homeostasis, bioenergetics and mitochondrial dynamics in neonatal rat cerebral cortex and striatum, brain structures that are most injured in HMGA patients [2,5,6].”

Lines 248-249 read that MFN1 is not statistically significantly changed but figure 5 shows that it is (*).

Response: We believe there is no (*) in the figure. Comparing the two bars (Control and HMG) in the graph, we can see a slight increase, but it is not statistically significant.

In the discussion section the increase found in CAT in the cortex and the decrease found in the striatum needs more detailed explanation.

Response: It is difficult to explain why HMG differentially affected the activity of catalase pending on the brain structure evaluated. We speculate that this effect might be dependent on the iron content since the striatum has higher levels of iron, rendering this brain structure more vulnerable to deleterious effects. The following sentences were added to the Discussion. 

“It is difficult to explain why HMG reduced CAT activity in the striatum but increased it in the cortex. However, it is widely known that the concentrations of iron, which catalyze the formation of reactive species [33], are higher in the striatum than in the cortex [34,35]. Thus, we speculate that the generation of ROS by HMG may occur through an iron-dependent mechanism. This hypothesis may also explain why the activity of many antioxidant enzymes, including CAT, was reduced in the striatum but not in the cortex.”

A reference for lines 332-335 should be provided.

Response: We inserted references for this sentence.

In lines 342-343: “In addition, we showed that HMG also caused oxidative stress and impaired the functioning of citric acid cycle in the cerebral cortex of 1-day-old rats albeit the effects were less pronounced [10]”.

Please rewrite this sentence. It is not clear. It appears that the authors want to discuss their results but they are referring to a work done by other authors.

Response: We thank the Reviewer for this observation. We have altered the sentence to better explain our point of view, as follows.

“On the other hand, previous data from our group showed that HMG also caused oxidative stress and impaired the functioning of citric acid cycle in the cerebral cortex of 1-day-old rats [10]. Although these effects in 1-day-old animals were less pronounced [10], this may be explained by the lower dose of HMG injected (0.5 mmol/g), compared to that used in the present study.”

Lines 346-347: If the authors state that no studies have measured the levels of HMG in the brain of patients at least they should provide information on how this disease is detected in these individuals.

Response: The levels of HMG in the brain of patients indeed have not been reported, but its presence has been confirmed (see Reference [1] of the manuscript). Therefore, we added some sentences in the manuscript to highlight this issue.

“It should be emphasized that even though HMG accumulation has been shown in the brain of patients through coupled brain and urine spectroscopy [1], no studies have so far reported the levels of this metabolite in this tissue.”

I also suggest that the inclusion of a simple diagram or scheme of the biochemical impaired pathway that leads to HMG accumulation would be useful.

Response: We created a figure showing the metabolic route with the biochemical deficiency.

Minor points:

Correct typo mistake line 52

Line 83: I suggest changing the word “moments” for a precise period as the “first year of life” or the perinatal period.

Line 183: please finish the sentence

Line 246: replace the doi for reference number.

Line 281: repetition of references.

Line 340: correct symbol in units.

Response: All minor points were corrected.

Best regards!

Reviewer 3 Report

Comments and Suggestions for Authors

This paper deals with 3-Hydroxy-3-methylglutaric acidemia (HMGA), a neurometabolic inherited disorder characterized by the predominant accumulation of 3-hydroxy-3-methylglutaric acid (HMG) in the brain and biological fluids of patients. Symptoms often appear in the first year of life and include mainly neurological manifestations. Since the neuropathophysiology is not fully elucidated the authors investigated the effects of intracerebroventricular administration of HMG on redox and bioenergetic homeostasis in the cerebral cortex and striatum of neonatal rats. The authors evaluated also some neurodevelopment parameters. The authors found that HMG decreased the activity of glutathione reductase (GR) and increased catalase (CAT) in the cerebral cortex. In the striatum, HMG reduced the activities of superoxide dismutase, glutathione peroxidase, CAT, GR, glutathione S-transferase, and glucose-6-phosphate dehydrogenase. Regarding bioenergetics, HMG decreased the activities of succinate dehydrogenase and the respiratory chain complexes II-III and IV in the cortex. HMG also decreased the activities of citrate synthase and succinate dehydrogenase as well as the complex IV in the striatum. HMG further increased DRP1 levels in the cortex, indicating mitochondrial fission. Finally, the authors found that the HMG-injected animals showed impaired performance in all sensorimotor tests examined. The authors concluded their study by stating that their findings provide evidence that HMG causes oxidative stress, bioenergetic dysfunction, and neurodevelopmental changes in neonatal rats that could explain the neuropathophysiology of HMGA.

The paper is potentially interesting however I believe that some limitations in the methods should be addressed by the authors.

-Did the authors use male or female rats?

-The authors state that the rats were euthanized by decapitation without anesthesia 6 hr after injection. However, neurodevelopment was evaluated 48 hr after injection (postnatal day 9-PND9). This is odd.

-The n of the two groups is hard to find in the paper.

-What is the rationale of the icv injection in 4th ventricle?

-F and dF should be included in the study

-How did the authors choose the HMG dose for icv administration (1 μmol/g)? This dose looks quite huge. Did they check for putative HMG side effects?

Comments on the Quality of English Language

 Minor editing of English language required

Author Response

This paper deals with 3-Hydroxy-3-methylglutaric acidemia (HMGA), a neurometabolic inherited disorder characterized by the predominant accumulation of 3-hydroxy-3-methylglutaric acid (HMG) in the brain and biological fluids of patients. Symptoms often appear in the first year of life and include mainly neurological manifestations. Since the neuropathophysiology is not fully elucidated the authors investigated the effects of intracerebroventricular administration of HMG on redox and bioenergetic homeostasis in the cerebral cortex and striatum of neonatal rats. The authors evaluated also some neurodevelopment parameters. The authors found that HMG decreased the activity of glutathione reductase (GR) and increased catalase (CAT) in the cerebral cortex. In the striatum, HMG reduced the activities of superoxide dismutase, glutathione peroxidase, CAT, GR, glutathione S-transferase, and glucose-6-phosphate dehydrogenase. Regarding bioenergetics, HMG decreased the activities of succinate dehydrogenase and the respiratory chain complexes II-III and IV in the cortex. HMG also decreased the activities of citrate synthase and succinate dehydrogenase as well as the complex IV in the striatum. HMG further increased DRP1 levels in the cortex, indicating mitochondrial fission. Finally, the authors found that the HMG-injected animals showed impaired performance in all sensorimotor tests examined. The authors concluded their study by stating that their findings provide evidence that HMG causes oxidative stress, bioenergetic dysfunction, and neurodevelopmental changes in neonatal rats that could explain the neuropathophysiology of HMGA.

The paper is potentially interesting however I believe that some limitations in the methods should be addressed by the authors.

-Did the authors use male or female rats?

Response: Male rats were used. We inserted this information in the Methodology.

-The authors state that the rats were euthanized by decapitation without anesthesia 6 hr after injection. However, neurodevelopment was evaluated 48 hr after injection (postnatal day 9-PND9). This is odd.

Response: The time point of 6 hr was chosen for the evaluation of biochemical paramters because previous studies from our group showed that different metabolites accumulating in inborn errors of metabolism elicit toxic effects at this or similar periods [Ribeiro et al. Neurotoxicity Research (2023) 41:119–140; Ribeiro et al., Neuroscience (2021) 471:115-132; Zemniaçak et al., Mol Neurobiol (2024) 61(5):2496-2513]. As for neurodevelopment, different reasons led us to evaluate it 48 hr after the injection: 1) we wanted to rule out a possible interference of the anesthesia on the markers; 2) the animals need a proper recovery time after HMG administration; 3) after most insults in animal models, changes in neurodevelopment cannot be detected after short periods. The following sentences were inserted in the text.

“These parameters were determined specifically on PND9 due to the following reasons: 1) to rule out a possible interference of the anesthesia; 2) to give an adequate recovery time after the injection; 3) to detect the possible neurodevelopment changes with more reliability since after most insults in animal models, such alterations cannot be detected after short periods.”

-The n of the two groups is hard to find in the paper.

Response: The N was corrected and can be found in the legends to figures.

-What is the rationale of the icv injection in 4th ventricle?

Response: We used the icv injection in the fourth ventricle because the surgery for intracerebral administration in neonatal rats cannot be performed in a stereotaxic apparatus due to their size. Moreover, an icv injection in neonatal Wistar rats is only possible in the fourth ventricle since it is performed manually and based on the position of bregma and lambda. In addition, given that we wanted to evaluate the effects of HMG in the cerebral cortex and striatum, two structures commonly affected in HMGA patients, we used the icv injection so the metabolite could reach both structures.

-F and dF should be included in the study

Response: We revised all statistical analyses and corrected some data. We also included F and dF in the results with statistical difference, as requested.

-How did the authors choose the HMG dose for icv administration (1 μmol/g)? This dose looks quite huge. Did they check for putative HMG side effects?

Response: We chose the dose of 1 μmol/g based on previous studies that evaluated the effects of HMG and other metabolites accumulating in different inherited metabolic disorders at similar [da Rosa et al., Neurotox Res (2020) 37: 314–325; Ribeiro et al., Neurotoxicity Research (2023) 41:119–140; Olivera et al., Neurobiology of Disease (2008) 32: 528–534]. It should be considered here that the brain concentrations of HMG in patients have not been reported yet. Furthermore, studies demonstrated that in cerebral organic acidurias, such as HMGA, the local production and entrapment of acidic compounds, in particular, dicarboxylates and acyl-CoA ester precursors, result in the pronounced accumulation of these compounds in the brain, causing toxic effects that lead to progressive neurodegeneration. Moreover, in such diseases, intracerebral synthesis and low efflux transport from the brain leading to the entrapment of organic acids have been proposed to participate in brain injury [Wajner, Nature Reviews-Neurology (2019) 15: 253-271]. Therefore, although we do not know the brain concentrations of HMG, it is conceivable that, in the intracellular milieu, this organic acid is found at very high levels. These reasons led us to use the relatively high dose of 1 μmol/g.

            Regarding the second question, we did not exactly evaluate HMG side effects, but the neurodevelopment markers shown in the manuscript were determined to investigate the consequences of HMG effects since patients have symptoms such as cognitive deficit and hypotonia.

Reviewer 4 Report

Comments and Suggestions for Authors

Authors have inaccuracy in affiliations, they should follow in the order 1,2,3,4,5. 

The text in lines 66-69 contains your research that was obtained in the present study. The place of this discovery of yours is in the discussion of the results you obtained.At the end of the introduction, you rush the discussion again. This paragraph has a place in the Discussion of Results chapter.

The standard deviations presented in the figures are unintelligible. How many standard deviations does each bar contain and what do they mean?

You have conducted experiments and obtained results that strongly demonstrate that HMG causes oxidative stress and bioenergetic dysfunction in the cerebral cortex and striatum of 7-day-old animals. In conclusion, however, it is not clear what this discovery of yours can serve. Can you say at this stage whether your results allow you to propose mastering this pathophysiology as a target for the prevention and treatment of the HMGA condition?

Comments on the Quality of English Language

The English language needs a little correction.

Author Response

Authors have inaccuracy in affiliations, they should follow in the order 1,2,3,4,5. 

Response: We corrected the affiliations.

The text in lines 66-69 contains your research that was obtained in the present study. The place of this discovery of yours is in the discussion of the results you obtained.At the end of the introduction, you rush the discussion again. This paragraph has a place in the Discussion of Results chapter.

Response: We believe that the Reviewer refers to the sentences describing the findings of a previous work from our group. We changed the sentences to clarify that, as follows:

“In the brain of 1-day-old rats, HMG further increased mitochondrial fusion and induced neural damage [9].” 

The standard deviations presented in the figures are unintelligible. How many standard deviations does each bar contain and what do they mean?

Response: We apologize for this. We corrected the standard deviation bars in all figures.

You have conducted experiments and obtained results that strongly demonstrate that HMG causes oxidative stress and bioenergetic dysfunction in the cerebral cortex and striatum of 7-day-old animals. In conclusion, however, it is not clear what this discovery of yours can serve. Can you say at this stage whether your results allow you to propose mastering this pathophysiology as a target for the prevention and treatment of the HMGA condition?

Response: It is difficult to extrapolate our results to the human pathological condition. However, we believe that our data identify mechanisms (oxidative stress and bioenergetic disturbances) that can be used as a target for novel therapeutic strategies. However, this interpretation must be taken with caution. We added the following sentence to the manuscript.

“Finally, our findings suggest that antioxidants, particularly those targeting mitochondria, and energy substrates could be considered potential therapeutic strategies for HMGA”.

Round 2

Reviewer 2 Report

Comments and Suggestions for Authors

I have revised the up dated version of the manuscript.

I suggest the authors to add a statement in the discussion section that there is no information whether there is a greater or lesser prevalence of one sex or another for this pathology.

I also suggest the authors to use of the same number of animals in terms of sex for future studies.

In my opinion, it is still necessary to provide data on how the scores of the behavioral tests were calculated so that the reader does not have to go to another work to find out how they are calculated. A very brief description can be made in the material and methods section as they did in a previous study:

Ribeiro, R.T.; Carvalho, A.V.S.; Palavro, R.; Durán-Carabali, L.E.; Zemniaçak, Â.B.; Amaral, A.U.; Netto, C.A.; Wajner, M. L-2-Hydroxyglutaric Acid Administration to Neonatal Rats Elicits Marked Neurochemical Alterations and Long-Term Neurobehavioral Disabilities Mediated by Oxidative Stress. Neurotox Res 2023, 41,  119–140, doi:10.1007/s12640-022-00625-0.

Reviewer 3 Report

Comments and Suggestions for Authors

The replies of the authors to most of my comments are fine.

However, the authors should include in the Discussion, as a study limitation, the issue of the administered HMG high dose including their explanation for the chosen dose.

Comments on the Quality of English Language

Minor editing of the English language required
